# Doped PANI Coated Nano-Ag Electrode for Rapid In-Situ Detection of Bromide in Seawater

**Qiujin Wang [1], Yifan Zhou [1], Jixue Zhou [2], Rongrong Wu [1], Jianbo Wu [1], Hao Zheng [1], Ying Ye [1] and Yuanfeng Huang [3,\*]**

[1]   Ocean College, Zhejiang University, Zhoushan 316000, China; 21734040@zju.edu.cn (Q.W.); zhouyifan0803@zju.edu.cn (Y.Z.); 21634038@zju.edu.cn (R.W.); oswien@zju.edu.cn (J.W.); zhenghao@zju.edu.cn (H.Z.); gsyeying@zju.edu.cn (Y.Y.)

[2]   Shandong Provincial Key Laboratory for High Strength Lightweight Metallic Materials, Advanced Materials Institute, Qilu University of Technology (Shandong Academy of Sciences), Jinan 250000, China; zhoujx@sdas.org

[3]   Shandong Special Equipment Inspection and Testing Science &Technology Co., Ltd., Jinan 250000, China

\*   Correspondence: huangyf@sdtj.sd.cn

**Abstract:** In this paper, we successfully fabricated a novel bromide ion selective electrode (Br-ISE), which was coated by bromine ion doped polyaniline as sensitive film. Using Ag wire as the substrate, a uniform and dense nano-silver layer was electroplated to enhance the specific surface area of the electrode. Subsequently, a polyaniline (PANI) film was coated onto the electrode by cyclic voltammetry in a 0.3 M aniline and 1 M HCl solution and was in-situ doped by 0.1 M KBr solution. The morphology and performance of the electrode were characterized by scanning electron microscopy (SEM), electrochemical impedance spectroscopy (EIS), and other electrochemical analysis methods, respectively. The prepared Br-ISE exhibited a wide linear dynamic range between $1.0 \times 10^{-1}$ and $1.0 \times 10^{-7}$ M with a near-Nernst slope of 57.33 mV/decade. In addition, the electrode possessed extremely fast response time (<1 s) and low impedance (300 $\Omega$), high sensitivity, and good selectivity. The electrode potential drifted within 2 mV in 8 h. The lifespan was larger than three months.

**Keywords:** nano-silver; electrode; bromide ion doped PANI; film; in-situ measurement

## 1. Introduction

Bromine is an element mainly found in seawater, underground concentrated brines, sedimentary rock deposits of ancient oceans, and salt lake waters [1]. About 99% of bromine in the earth exists in the form of $Br^-$ in seawater [2], so bromine is also called "a marine element". The concentration of bromide needs to be determined during the extraction process of bromine from seawater, such as the air blowing method [3], ion exchange resin method [4], and steam distillation method [5]. China produces the third highest amount of bromine in the world, with air blowing being the most widely used in method. It is fundamental to monitor the bromine content in real-time during bromine extraction in order to calculate the chlorine distribution rate. However, the bromine industry normally use titration to measure the bromine concentrations. Methods to detect the concentration of bromide and other halogens in water include neutron activation analysis, inductively coupled plasma mass spectrometry, ion and liquid chromatography, and fluorescence spectroscopy [6–11]. Although these methods are highly mature and could obtain results with high accuracy and precision, complex procedures and expensive equipment limited these methods. In addition, it is difficult to integrate online devices using these methods, and some reagents will be introduced during measurement, which may cause secondary environmental pollution.

Liquid junction ion-selective electrode is another mature and widely used method to determine the bromine concentrations in water. However, conventional liquid membrane ion selective electrode method suffers from signal drift as a result of the diffusion of the internal electrolyte through the sensitive membrane [12]. At the same time, it is difficult to minimize and preserve the electrodes due to the existence of internal reference solutions. When compared with the traditional liquid junction ion selective electrode, all-solid-state ion-selective electrode shows the advantages of having a small size, being unaffected by external pressure and temperature, and being easy to store and integrate. They have become an important research frontier of ion-selective electrodes. The most widely studied all-solid-state ion-electrodes are comprised of conductive polymer materials, such as polypyrrole (PPy) [13], poly-3-octylthiophene (POT) [14], polyaniline (PANI) [15–19], poly(3,4-ethylenedioxythiophene) (PEDOT) [20,21], and so on. In addition, inorganic membrane electrodes are mostly insoluble salt electrodes [22], such as Ag/AgCl reference electrode [23], $Ag/Ag_2S$ electrode [24], and $Ir/IrO_x$ pH electrode [25]. Organic and inorganic composite membranes have been used to fabricate electrodes [26].

Our research group has successfully fabricated various all-solid-state ion-selective electrodes that can be applied to detect aqueous ions, such as ammonium ions [27], sulfate [28], sulfur ions [29], and carbonates [30]. Sljukić [31] coated three types of nanoparticles (silver, gold, and palladium) onto the surface of glassy carbon spherical powder by electrodepositing for bromine ion-selective electrodes, and these three materials were then combined together onto a basal plane pyrolytic graphite electrode surface and a combined approach to finding the electrode material for bromide detection as model target analyte was applied. Milikić [32] synthesized silver (Ag) and four carbon-supported Ag catalysts by the γ-radiation, and all five Ag materials exhibited electroactivity for the sensing of $Br^-$, with pure Ag catalyst giving the best response to $Br^-$ presence; while, Nyachhyon at al. [33] reported a bromine ion selective electrode (Br-ISE) that is composed of polycrystalline $Ag_2S$–AgBr film by the co-precipitation of silver sulfide and silver bromide; and some researchers [34–38] fabricated a Br-ISE that is based on PVC material synthesized ionophore.

Nevertheless, Br-doped PANI was rarely used as the bromide-sensitive film for Br-ISE. In this paper, we prepared a new all-solid-stage Br-ISE coated by bromide ion doped PANI electropolymerized film that was was prepared using cyclic voltammetry. We determined the working range, response time, and lifespan of the prepared Br-ISE. In addition, we applied the Br-ISE in a practical industry (Shandong Haihua Group Co., Ltd.) to in-situ monitor the bromine contents of seawater.

## 2. Experimental

### 2.1. Reagents and Apparatus

Ag wire (99.99%, 0.5 mm in diameter) was purchased from the Precious Materials Company of Changzhou, Changzhou, China. Sodium bromide, aniline, hydrochloric acid, sodium chloride, sodium sulfite, sodium sulfate, sodium nitrate, acetone nitric acid, and silver nitrate were obtained from the Aladdin Company (Shanghai, China). All of these chemicals were of analytical grade. The water used to configure the solution was ultrapure. The fabrication and electrochemical analyses were carried out on an electrochemical workstation (IviumStat, Ivium Technologies B.V. Company, Eindhoven, The Netherlands). Three-electrode-system was applied in the processes: the auxiliary electrode was a 10 mm × 10 mm × 0.1 mm Pt (99.99%) electrode (Shanghai Chenhua Instrument Co., Ltd., Shanghai, China) and the reference electrode was an Ag/AgCl reference electrode (CHI111, Shanghai Chenhua Instrument Co., Ltd., Shanghai, China). A FLUKE 123B Industrial Scope meter (Fluke Testing Instruments (Shanghai) Co., Ltd., Shanghai, China) and a SG102A function signal generator (Ruite Company, Suzhou, China) were used to generate a half sine wave voltage. A CNC ultrasonic cleaner KQ218 (Kunshan Ultrasonic Instrument Co., Ltd., Suzhou, China) was used to clean the Ag wires and accelerate dissolution. Scanning electron electrode microscopy (SU-8010, Hitachi, Tokyo, Japan) was used to observe the surface morphology of the Ag-coated electrodes.

### 2.2. Preparation of Nano Silver

There are many methods for preparing nano-Ag materials, which can be summarized as physical methods, chemical methods, and radiation methods. The electrochemical methods can be carried out at normal temperature and pressure without the need for subsequent processing, which saves energy [39]. Metal electrodeposition is a kind of electrochemical method that refers to the reduction reaction of metal ions on the conductive solid surface by the action of electric current, thus forming metal atoms on the solid surface. In this paper, the function signal generator supplies power for the circuit [40]. The Schottky diode has high switching frequency and low forward voltage, which plays a rectifying role in the circuit. Nano-silver coating was obtained on the surface of cathode silver wire with a 0.1 M concentration of $AgNO_3$ solution as electrolyte by setting the frequency (50 Hz) and amplitude of the current (800 mV) generated by the function signal generator and choosing the reasonable signal wave type.

### 2.3. Bromide Ion Doped PANI Film

Silver/nano-silver was used as a carrier for the electrochemically polymerize PANI film with a certain thickness. By using an electrode embryo as the working electrode, a Ag/AgCl electrode as the reference electrode, and a Pt as the auxiliary electrode, the cyclic voltammograms of all three electrodes were scanned for 10 cycles at the scan rate of 100 mV/s in an aqueous solution of 1.0 M HCl and 0.3 M aniline at the potential range of −0.2 to 0.9 V. After that, the prepared electrode was rinsed with response alcohol to remove the residual aniline oligomer on the surface [41]. Transferring the three electrodes into a 0.1 M KBr solution and scanning for another 20 cycles at the potential range of −0.2 to 0.72 V obtained the sensitive film that is based on the above product. It is important to note that, when PANI is electropolymerized in an aqueous solution of 1.0 M HCl + 0.3 M aniline, $Cl^-$ in the solution was doped into the main chain of PANI [42,43], whereas the $Br^-$ replace the $Cl^-$ to form a Br-doped PANI film covering the surface of Nano-Ag (as shown in Figure 1). This result facilitates the stable and fast response of the electrode to aqueous $Br^-$.

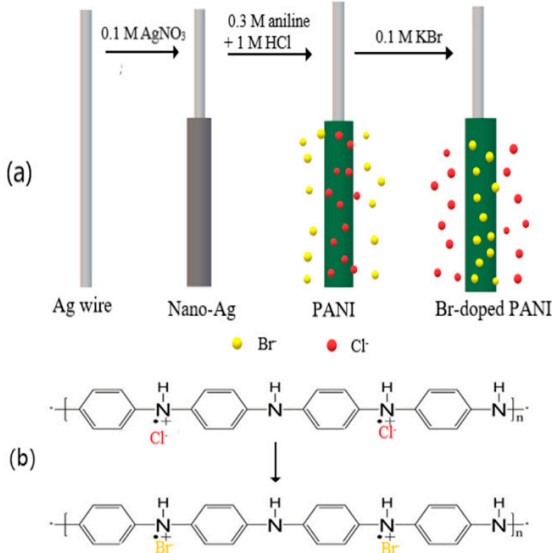

**Figure 1.** Schematic diagram of electrode fabrication steps (**a**) and molecular structure of chloride-doped (**b**).

### 2.4. Characterization

Scanning electron electrode microscopy (SU-8010, Hitachi) was used to observe the micromorphology of the nano-Ag layer, the PANI layer, and Br-doped PANI layer. The elemental composition of the Br-doped PANI layer was determined by energy-dispersive spectroscopy (SU-8010, Hitachi). In addition, the EIS of the Br-ISE was measured in 0.01 M KBr solution while using a

three-electrode system with an operating potential of 10 mV (open circuit voltage) and a frequency range of 0.1 Hz to 1 MHz (by IVIUM electrochemical workstation). The impedance data was analyzed by ZView 2. The Drop Snake plugin of ImageJ 1.52o was used to measure the contact angle.

The Br-ISE was calibrated in the KBr ($10^{-6}$–$10^{-1}$ M) solutions that were prepared in deionized water for 200 s. The effect of the pH value of the solution on the Br-ISE response potential was studied by calibrating the potential of Br-ISE in $10^{-3}$ M KBr solutions with different pH values (2–13). To measure the response time and sensitivity of the Br-ISE, the prepared electrode was calibrated for 200 s in $10^{-6}$–$10^{-1}$ M KBr solution without any pretreatment with a sample interval set to 1s. All of the potentiometric measurements were obtained with the IVIUM workstation at room temperature.

The reproducibility, repeatability, and the lifespan of the electrode are important factors in determining the large-scale production and practical application of ISEs. Eight electrodes were prepared under the same conditions using the same fabricating method to verify the repeatability of the electrodes. In life testing of the Br-ISE, the Br-ISE was stored in $10^{-3}$ M KBr solution for 90 days and regularly calibrated in $10^{-6}$–$10^{-1}$ M KBr solutions.

Selectivity is one of the most important performance indicators of a sensor, and it usually determines whether the test is reliable. In this paper, the FIM (fixed interference method), as recommended by the International Union of Theoretical and Applied Chemistry (IUPAC), is used to determine the selectivity of interfering ions of bromide electrodes.

$$K_{i,j}^{\text{pot}} = \frac{a_i}{a_j^{n_i/n_j}} \tag{1}$$

where $i$ is the measured ion and $j$ is the interference ions. $a_i$, $a_j$ are the activities of the measured ion and interference ion, respectively, and $n_i$, $n_j$ are the number of charges of the measured ion and interference ion, respectively.

## 3. Results and Discussion

### 3.1. Electro-Polymerization of PANI and Bromine Ion Doped PANI

As shown in Figure 2, there are two pairs of noticeable redox peaks near 0.2 and 0.7 V, which are identical to the report values [44]. The peak at 0.2 V correlates with the process of aniline oxidation of cation ion groups and the peak at 0.7 V correlates to the further oxidation of quinone compounds. The peak current decreased with continuous potential scanning, which indicates that the polymer film was successfully deposited on the surface.

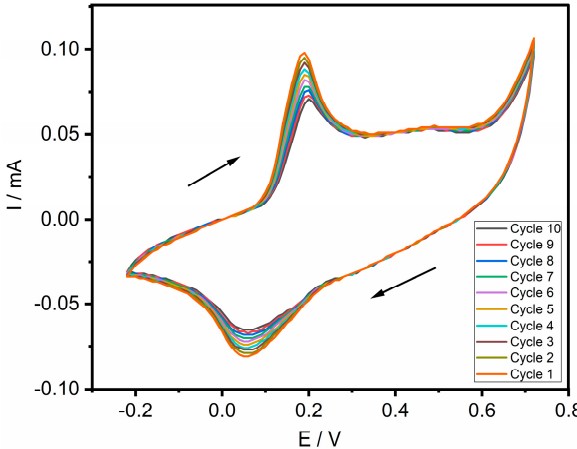

**Figure 2.** Electroplating polyaniline (PANI) film in an aqueous solution of 1.0 M HCl + 0.3 M aniline by cyclic voltammetry.

Figure 3 reflected the process of bromide ion doped PANI, which is similar to the CV curve when polymerizing aniline, the oxidation peak at 0.2 V was enhanced, while the oxidation peak at 0.7 V was weakened. In addition, the reduction peak is enhanced, which suggests that the oxidation peak at 0.2 V during electro polymerization was not only related to the aniline oxidation of aniline to a cation ion groups, but also connected to the doping of bromine ions (since there is only Br$^-$ in the solution and no aniline monomer). The redox peak at 0.7 V was the polymerization peak of aniline [45].

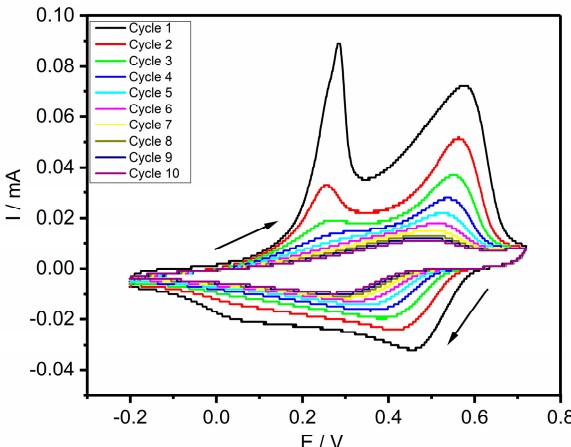

**Figure 3.** Doped PANI in 0.1 M KBr Solution by cyclic voltammetry.

### 3.2. Scanning Electron Microscopy (SEM), Energy-Dispersive Spectroscopy (EDS), and Contact Angle Measurement

The Figure 4 shows the SEM images of silver particles that were coated on the surface of the silver wire. The silver particles are evenly distributed on the surface of the substrate (Figure 4a). Figure 4b shows that silver particle size is on the nanometer scale. The silver particles increased the specific surface area of the electrode, which facilitates the polymerization of the PANI layer. Figure 5 shows the micro-morphology of PANI and the bromine ion-doped PANI polymer. The PANI film appeared as granular aggregate, with diameters of about 200 nm in diameter, and the entire coating was uniform and dense (Figure 5a). At the beginning of doping, the electropolymerized PANI film was an electron mediator transfer electrons and, simultaneously, a site to dope bromide ions at the same time [46,47]. Doping changed the morphology of the polymerized film, which resulted in fine conductive channels and bromide recognition sites (Figure 5b). The water contact angle of the Br-doped PANI film is 59.8° (Figure 5c), which indicates that the coating is hydrophilic. Wettability affects the performance of the electrode: greater water affinity contributes to the higher sensitivity and shorter response time of the electrode.

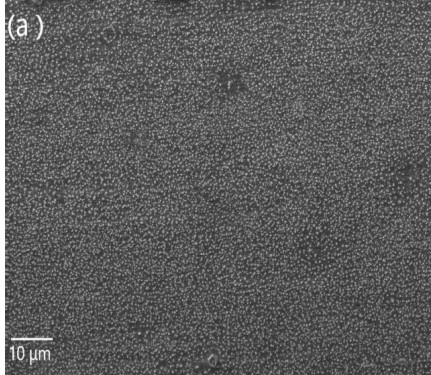
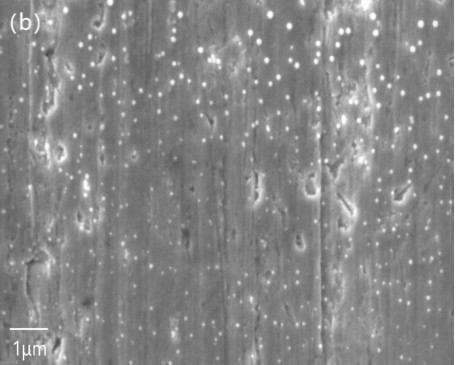

**Figure 4.** The Scanning electron microscopy (SEM) images of nano-silver after in magnification of 1000× (**a**) and 8000× (**b**).

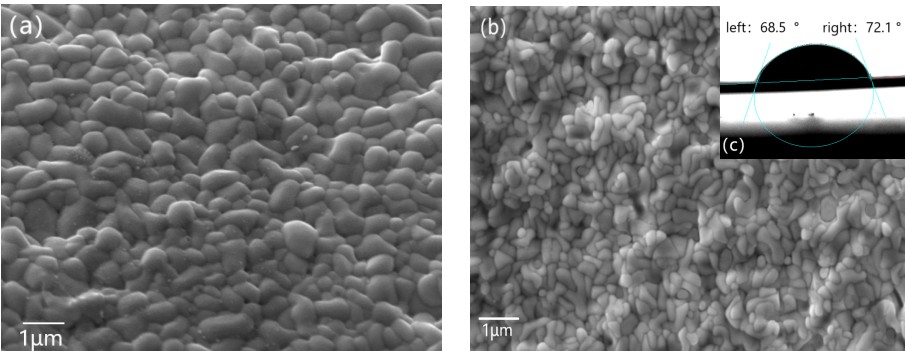

**Figure 5.** The SEM images of PANI with a magnification of 8000 (**a**) and bromine ion doped PANI (**b**); The water contact angle of the Br-doped PANI film (**c**).

Energy-dispersive spectroscopy (EDS) determined the elemental composition of the Br-doped PANI layer. When PANI was electroplated on the silver wire, a large number of chloride ions were grafted onto the main chain of the PANI. After doping with bromide ions, the chloride ions were replaced by bromine ions, and the presence of Br suggested that the Br-doping was successful (Figure 6).

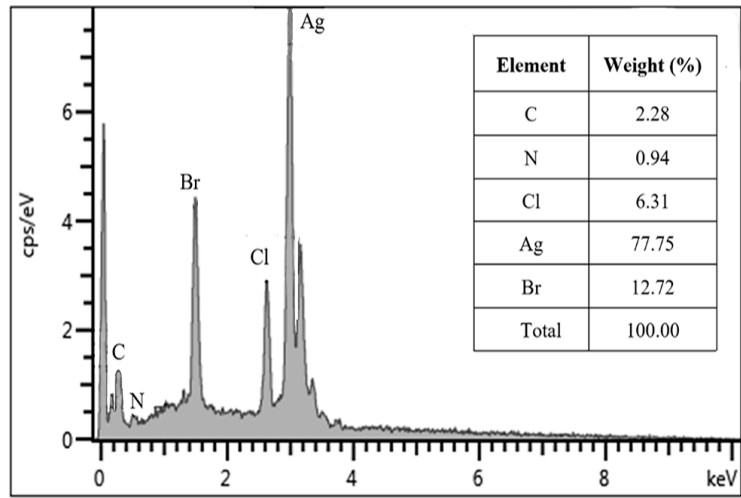

| Element | Weight (%) |
|---------|-----------|
| C | 2.28 |
| N | 0.94 |
| Cl | 6.31 |
| Ag | 77.75 |
| Br | 12.72 |
| Total | 100.00 |

**Figure 6.** The elemental analysis of Br-doped PANI.

### 3.3. Electrochemical Impedance Spectroscopy (EIS)

The internal structure of the electrode sensitive film can be simulated as "components" or "layers" in an electrical equivalent circuit (ECC). The traditional glass electrode gennerally shows a stable response and good repeatability, but its higher impedance limits it ($10^8$–$10^{12}$ Ω) [48]. The symbols that are used in this circuit have the following meaning: $R_s$ is solution resistance and $R_{ct}$ is charge transfer resistance, $C$ is the double layer capacitance at the metal electrolyte interface, and $W_s$ is the Warburg impedance. As shown in Figure 7, the impedance of the Br-ISE is as low as 300 Ω. Low impedance indicates better conductivity and higher measurement accuracy. The low impedance of the prepared Br-ISE was a result of the doping of Br$^-$, which made the surface of the electrode denser (Figure 5). When AC signals pass through the electrodes, the diffusion control exceeds the electrochemical control at low frequencies (LF), which results in Warburg impedance [49,50].

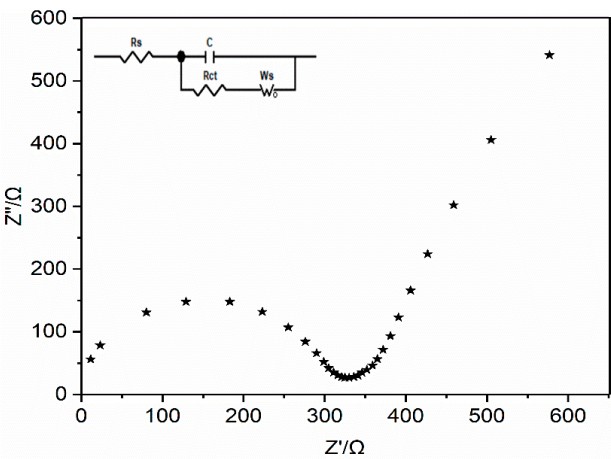

**Figure 7.** Nyquist plots of Br-doped PANI.

### 3.4. Electrode Calibration, Nernst Response, Linear Range, and Detection Limit in Freshwater

The Nernst slope of the two electrodes are 57.28 and 58.82 mV/decade, respectively, close to the calculated theoretical value of 59.20 mV/decade. Operating errors during fabrication caused the slight difference between the two electrodes. During the calibration process, the electrode presented good stability with the potential drift ≤2 mV, and high regression coefficients of the Nernst equation ≥0.99.

### 3.5. Response Time and Stability

As shown in the Figure 8, there no obvious potential drift was observed during the whole calibration process. The red part in the Figure 9 is the calibration data in the first 20 s. It can be clearly seen from the data that, within 1 s, the electrode quickly enters a stable state, so the response time of the electrode is less than 1 s.

Figure 10 shows the results of the stability test of the bromide ion electrode. The lowest potential was 41.06 mV, the highest potential was 42.56 mV, the potential drift was less than 1.3 mV, and the average potential drift was 0.16 mV per hour. The potential drift mainly occurred during the first 5 h. In the last 3 h, the potential started to stabilize, and it resulted in the potential drift of less than 0.72 mV. Therefore, the prepared electrode can potentially be used for long-term online monitoring because of its good stability.

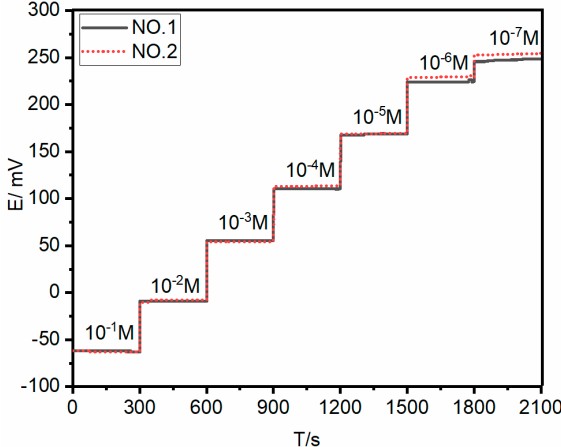

**Figure 8.** The calibration process of two electrodes (Nos. 1 and 2) in KBr prepared in freshwater.

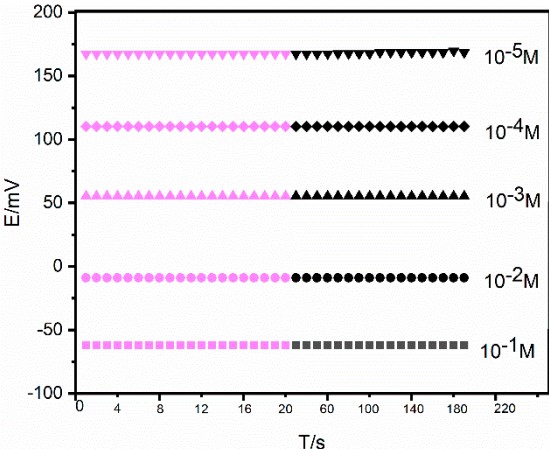

**Figure 9.** Response time of bromine ion selective electrode (Br-ISE) in different concentrations of KBr solutions.

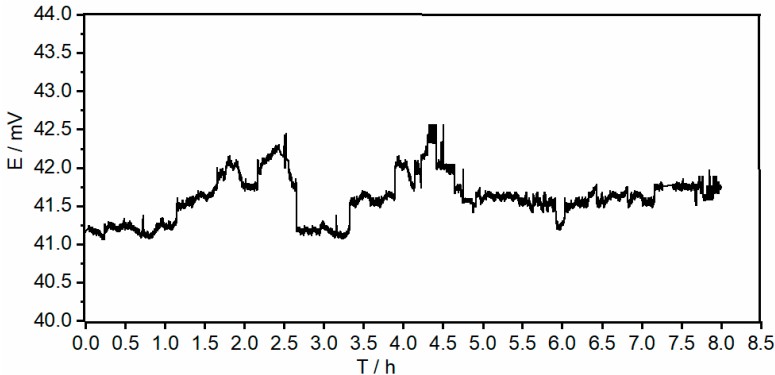

**Figure 10.** The stability of the Br-ISE was continuously calibrated in $10^{-3}$ M KBr solution for 8 h.

## 3.6. Effect of pH and Selectivity

The result that is presented in Figure 11 indicates that the potential response remained constant at the pH range of 2.0–10.0, which can be used as the applicable pH range of the prepared electrode. At pH <2.0, the response potential of bromide electrode significantly decreased with decreasing pH values due to the simultaneous response of the electrode to the oppositely charged $H_3O^+$ and $Br^-$ [51]. When the pH is greater than 10, on the contrary, $OH^-$ strongly competed with $Br^-$ to form oxonium cations, which resulted in an augmented response potential with increasing pH values [52].

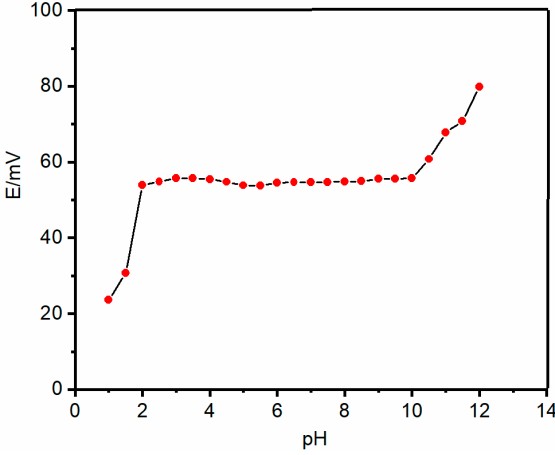

**Figure 11.** Effect of pH on potential response of the Br-ISE using $1 \times 10^{-3}$ M KBr solution.

The main interfering ions that are selected (including $Cl^-$, $I^-$, $F^-$, $NO_3^-$, $NO_2^-$, $SO_4^{2-}$, $SO_3^{2-}$, $SCN^-$) were fixed at $10^{-2}$ M (see Table 1). When compared with previous works, our prepared Br-ISE showed the lowest the selectivity coefficients for $Cl^-$. The selectivity of the as-prepared electrode for other ions, including $I^-$, $F^-$, $NO^{3-}$, $SO_4^{2-}$, $SO_3^{2-}$, and $SCN^-$ were also generally lower than the previous works. Therefore, it is indicated that the as-prepared Br-ISE possesses superior bromide ion-selectivity over interference ions.

**Table 1.** Selectivity coefficient of various interfering anions.

| Ref. No. | $Cl^-$ | $I^-$ | $F^-$ | $NO_3^-$ | $NO_2^-$ | $SO_4^{2-}$ | $SO_3^{2-}$ | $SCN^-$ |
|---|---|---|---|---|---|---|---|---|
| 19 | $8 \times 10^{-2}$ | 3.98 | – | – | – | – | – | – |
| 20 | $1.2 \times 10^{-2}$ | $1.8 \times 10^{-3}$ | $2 \times 10^{-5}$ | $2 \times 10^{-5}$ | $1 \times 10^{-5}$ | $1.5 \times 10^{-5}$ | $2 \times 10^{-5}$ | $1 \times 10^{-3}$ |
| 21 | $8.6 \times 10^{-2}$ | $6.4 \times 10^{-2}$ | $4.2 \times 10^{-5}$ | $8.4 \times 10^{-4}$ | $1.2 \times 10^{-5}$ | $3.2 \times 10^{-4}$ | – | $1 \times 10^{-3}$ |
| 22 | $1.4 \times 10^{-3}$ | $5.5 \times 10^{-3}$ | $7.5 \times 10^{-4}$ | $8.5 \times 10^{-4}$ | $8 \times 10^{-4}$ | $3 \times 10^{-4}$ | $3.1 \times 10^{-4}$ | $7.5 \times 10^{-3}$ |
| 23 | $9.0 \times 10^{-4}$ | $1 \times 10^{-3}$ | $2.6 \times 10^{-5}$ | $8 \times 10^{-3}$ | $4 \times 10^{-3}$ | $2 \times 10^{-4}$ | – | $3 \times 10^{-3}$ |
| 24 | $7.9 \times 10^{-2}$ | $2 \times 10^{-2}$ | – | $8.9 \times 10^{-2}$ | $6 \times 10^{-3}$ | $3 \times 10^{-3}$ | – | $6.9 \times 10^{-2}$ |
| This work | $4.7 \times 10^{-4}$ | $3.7 \times 10^{-3}$ | $6.8 \times 10^{-5}$ | $3.4 \times 10^{-4}$ | $5.2 \times 10^{-4}$ | $4.3 \times 10^{-5}$ | $6.5 \times 10^{-5}$ | $7.3 \times 10^{-3}$ |

### 3.7. Reproducibility, Repeatability and Lifespan

As shown in Figure 12, all the eight electrodes show comparable calibration curves. The Nernst slopes of these electrode were about 55 mV/decade, and the linear correlation coefficients were greater than 0.99. Therefore, it is indicated that our Br-ISE is easily reproducible. In addition, the Br-ISE is calibrated for nine times at different times, while its Nernst slope remained relatively unchanged, which suggested the good repeatability of the electrode.

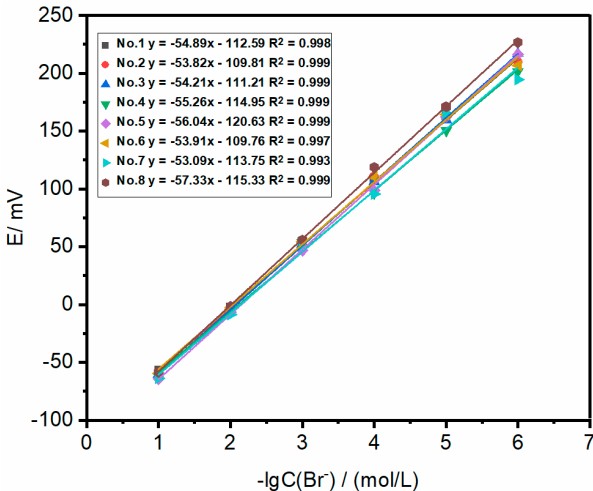

**Figure 12.** The calibration curves of eight Br-ISEs that prepared by using the same method.

Table 2 shows the slopes and the correction coefficients $R^2$ of the electrode. The slopes varied slightly stable between $-56.46$ mV/decade and $-54.41$ mV/decade with $R^2 > 0.99$, indicating that the lifespan of the electrode can reach 90 days.

**Table 2.** Calibration result of Br-ISE in 90 days.

| Days | Slope | Correlation Coefficient ($R^2$) |
|---|---|---|
| 1 | −56.04 | 0.9992 |
| 3 | −56.46 | 0.9952 |
| 8 | −55.36 | 0.9929 |
| 21 | −56.02 | 0.9911 |
| 36 | −55.62 | 0.9954 |
| 50 | −54.76 | 0.999 |
| 68 | −54.54 | 0.9958 |
| 83 | −55.56 | 0.9993 |
| 90 | −54.41 | 0.9997 |

## 4. Application

　　Our prepared electrodes were tested in the practical bromine production lines in Shandong Haihua Co., Ltd. (Weifang, China) and Shandong Yanye Co., Ltd. (Weifang, China). The industrial plant is a bromine factory that is located in the northeast of Shouguang City, China, and it borders on Laizhou Bay in the Bohai Sea. The production line was established in 1971 and was the first domestic company to use air blowing to produce bromine. At present, it has 13 bromine production lines with an annual production capacity of over 5,000 t. Caiyangzi bromine factory of Shandong Yanye Co., Ltd. is located in the southwestern coast of Laizhou Bay in the Bohai Sea, north of Shouguang City. It was founded in 1959 and contains four billion tons of high concentration brine water, which provides abundant resources for bromine production.

　　In our monitoring system, the Br-ISE was incorporated into the independently developed multi-parameter electrochemical sensor, as presented in Figure 13, and the data that was collected in real time was incorporated into the DCS/PLC system to realize the real-time monitoring of bromine ions and other parameters. The brine contains $MgSO_4$, $MgCl_2$, $Ca(HCO_3)_2$, $KCl$, $CaCl_2$, $NaCl$, etc. It is known that $Cl^-$ has is the strongest interfering effect to the Br-ISE from the selectivity results of the electrode (in Table 2). Therefore, the Br-ISE needs to be calibrated before the measurement. The chloride ion content in the Shandong Laizhou Bay brine is about 107 g/L. Hereby, we use $1 \times 10^{-1}$, $5 \times 10^{-1}$, $1 \times 10^{-2}$, $5 \times 10^{-2}$, $1 \times 10^{-3}$, $5 \times 10^{-3}$, and $1 \times 10^{-4}$ M KBr solutions with 10% NaCl to simulate the calibration of bromide ions in brine. Table 3 shows the real-time monitoring data of No. 1 and No. 2 Br-ISE within 12 h. $E_1$ and $E_2$ represent the potential values monitored by No.1 and No.2 Br-ISE in the brine and $C_1$, $C_3$ are the concentration values calculated from $E_1$, $E_2$, $C_2$, and $C_4$ are, respectively, and measured by on-site manual sampling using the traditional sodium hypochlorite-iodine method. The accuracy of the electrode could be measured by the actual change ($\Delta$, the data that $C_1$, $C_3$ are being compared to). Calculated from Table 3, the maximum $\Delta$ for Caiyangzi and Haihua are 3.92 and 2.29 mg/L. Moreover, the average percentage change ($\Delta$%) for the two factories are only 1.79% and 0.66%, respectively. Thus, our in-situ electrodes fully meet the demands of bromine plant for real-time monitoring of the bromine content in brine reservoir, which will save on labor costs and facilitate achieving industrial upgrading.

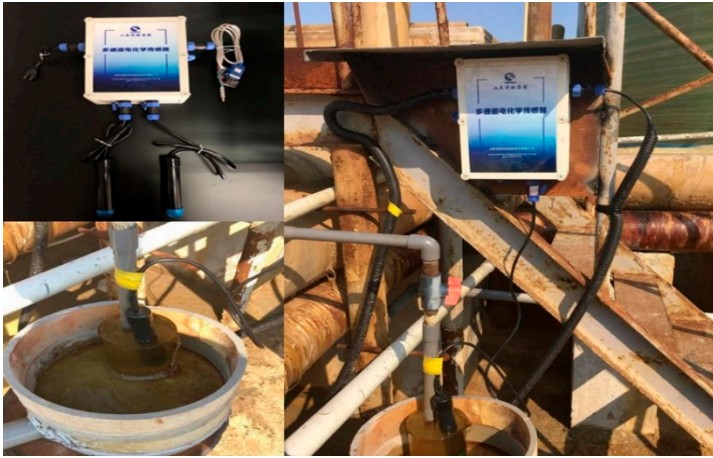

**Figure 13.** Multi-parameter electrochemical sensor working in the factory.

**Table 3.** On-site real-time monitoring data of bromide electrodes.

| Time Monitoring Site | Caiyangzi Bromine Factory | | | Haihua Group Bromine Factory | | |
| | Data by Br-ISE (No. 1), Nernst Equation $y = -48.497x - 108.57$ | | Data by Artificial | Data by Br-ISE (No. 2), Nernst Equation $y = -49.446x - 110.44$ | | Data by Artificial |
| | $E_1$ (mV) | $C_1$ (mg/L) | $C_2$ (mg/L) | $E_2$ (mV) | $C_3$ (mg/L) | $C_4$ (mg/L) |
|---|---|---|---|---|---|---|
| 8:00, Mar. 2nd, 2018 | 15.62 | 143.21 | 147.13 | 21.01 | 170.07 | 170.42 |
| 9:00, Mar. 2nd, 2018 | 15.31 | 145.52 | 147.45 | 21.23 | 168.30 | 170.47 |
| 10:00, Mar. 2nd, 2018 | 15.32 | 145.52 | 147.53 | 21.04 | 169.83 | 170.44 |
| 11:00, Mar. 2nd, 2018 | 15.41 | 144.75 | 147.60 | 21.21 | 168.46 | 170.47 |
| 12:00, Mar. 2rd, 2018 | 15.41 | 144.75 | 147.58 | 21.05 | 169.75 | 170.32 |
| 13:00, Mar. 2nd, 2018 | 15.60 | 143.21 | 146.90 | 21.12 | 169.18 | 170.38 |
| 14:00, Mar. 2nd, 2018 | 15.30 | 145.52 | 146.87 | 20.95 | 170.56 | 170.64 |
| 15:00, Mar. 2nd, 2018 | 15.32 | 145.52 | 146.78 | 20.89 | 171.04 | 170.54 |
| 16:00, Mar. 2nd, 2018 | 15.64 | 143.21 | 146.60 | 20.93 | 170.72 | 170.53 |
| 17:00, Mar. 2nd, 2018 | 15.63 | 143.21 | 146.79 | 21.15 | 168.94 | 170.65 |
| 18:00, Mar. 2nd, 2018 | 15.65 | 143.21 | 146.77 | 20.76 | 172.10 | 170.78 |
| 19:00, Mar. 2nd, 2018 | 15.42 | 144.75 | 146.66 | 21.22 | 168.38 | 170.67 |
| 20:00, Mar. 2nd, 2018 | 15.45 | 144.75 | 146.69 | 21.16 | 168.86 | 170.54 |

## 5. Conclusions

In this paper, a micro, all-solid-state and high sensitivity bromide ion selective electrode was prepared by the electrochemical method. A smooth and dense Nano silver layer was coated on the surface of the smooth silver wire substrate to reduce the impedance of Br-ISE. The electrode achieved better performance characteristics by doping the PANI backbone with bromide ions. The micrographs of scanning electron microscopy show that the bromide ion doping polymer film was uniform and dense without fine pores. The water contact test showed a good hydrophilicity on the surface, which contributes to the good sensitivity and selectivity of the electrode. Table 4 lists the characteristic comparison of Br-ISE between the previous works and our work. The electrode possessed a linear operating range of $10^{-7}$–$10^{-1}$ M and response time of less than 1 s. The potential drift was only 0.16 mV per hour in the long-term stability test. Additionally, and importantly, the electrode have long lifespan of more than three months and good selectivity over $Cl^-$ and $I^-$. The bromide ion electrode is being assembled with other electrodes into a multi-parameter electrochemical sensor to realize the real-time monitoring of various parameters in the water bodies.

**Table 4.** Performance comparison of bromide electrode between previous works and current work.

| Ref. No. | Slope (mV/Decade) | Linear Range (M) | Detection Limit (M) | Respond Time (s) | pH Range | Life Span (Weeks) |
|---|---|---|---|---|---|---|
| 19 | 58.0 | – | $1 \times 10^{-5}$ | – | – | – |
| 20 | 61.0 | $1 \times 10^{-2}$–$1 \times 10^{-5}$ | $4 \times 10^{-6}$ | – | 4.0–8.3 | – |
| 21 | 59.2 | $1 \times 10^{-1}$–$2.2 \times 10^{-6}$ | $1.4 \times 10^{-6}$ | 20 | 3.5–9.5 | 12 |
| 22 | 59.0 | $1 \times 10^{-1}$–$7 \times 10^{-6}$ | $6 \times 10^{-6}$ | ≤15 | 3.0–9.0 | 9 |
| 23 | 59.1 | $1 \times 10^{-1}$–$1 \times 10^{-5}$ | – | ≤20 | 4.0–9.5 | 8 |
| 24 | 61.0 | $1 \times 10^{-1}$–$3.2 \times 10^{-5}$ | $2 \times 10^{-5}$ | – | 4.5–8.5 | – |
| This work | 59.2 | $1 \times 10^{-1}$–$1 \times 10^{-7}$ | $6.29 \times 10^{-6}$ | <1 | 2.0–10.0 | 12 |

**Author Contributions:** Conceptualization, Q.W., Y.H. and Y.Y.; Methodology, Y.H.; Software, Q.W.; Validation, Y.Z. and J.Z.; Formal Analysis, Y.H.; Investigation, J.Z.; Resources, Y.Y.; Data Curation, Q.W. Y.Z.; Writing—Original Draft Preparation, Q.W.; Writing—Review and Editing, Y.H., R.W. and J.W.; Visualization, J.Z.; Supervision, H.Z.; Project Administration, H.Z.; Funding Acquisition, H.Z. and Y.Y.

**Funding:** This research was funded by the National Natural Science Foundation of China (NSFC No. U1709201).

**Acknowledgments:** The authors acknowledge the researchers at in Qilu University of Technology (Shandong Academy of Sciences) for their valuable discussions during this study.

**Conflicts of Interest:** The authors declare no conflict of interest.

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
