# Peer review of "Doped PANI Coated Nano-Ag Electrode for Rapid In-Situ Detection of Bromide in Seawater"

_coatings, doi:10.3390/coatings9050325_

Round 1

Reviewer 1 Report

The manuscript coatings-490593 entitled “Doped Polyaniline coated Nano-Ag electrode for rapid in-situ detection of bromide in seawater” presents a research work on an ISE for bromide ion based on doped polyaniline. Although the topic is interesting, the manuscript has weaknesses that must be addressed before its publication:

1. Please, revise the English throughout the manuscript.

2. Table 1. The comparison of the obtained analytical characteristics with the previously published sensors is not part of the “Introduction” Section. Therefore, I recommend to move the Table 1 and the related comments to the Discussion or the Conclusions.

3. Figure 3 and related comments. As far as I know, during the conditioning step with Br-, no electrochemical reaction occurs, just an interchange of the counter ion inside the polymer: Cl- by Br-. Then, what is the reason of performing a CV?. And what are the electrochemical processes related with the different peaks of this CV?. Please, clarify this important point.

4. Figures 7 and 8 are interchanged. Be careful.

5. I miss statistical parameters, such as a coefficient of variation (CV (%)) to evaluate the repeatability (the same sensor, calibrate different times) and reproducibility (different sensors) of this new ISE.

6. The authors claim in the Abstract and in the Conclusions that the developed ISE has a long lifetime of more than 3 months… but there is no experimental data to prove it!. Please, provide results to the main text.

Author Response

Point 1: Please, revise the English throughout the manuscript.

Response 1: My manuscript has been checked by professional English editing service at https://www. mdpi.com/authors/English.

Point 2: Table 1. The comparison of the obtained analytical characteristics with the previously published sensors is not part of the “Introduction” Section. Therefore, I recommend to move the Table 1 and the related comments to the Discussion or the Conclusions.

Response 2: We have removed the Table 1 and the related comments to the Conclusions.

Point 3: Figure 3 and related comments. As far as I know, during the conditioning step with Br, no electrochemical reaction occurs, just an interchange of the counter ion inside the polymer: Cl by Br-. Then, what is the reason of performing a CV? And what are the electrochemical processes related with the different peaks of this CV? Please, clarify this important point.

Fig.1 Doped polyaniline in 0.1 M KCl Solution by cyclic voltammetry.

Response 3: We added references [4243] in line 130 and explanations for different peaks. In addition, because polyaniline doping and cyclic voltammetry are complex, a supplementary experiment proves that the peak in Fig.3 is indeed generated by doping from the opposite side. The other steps are the same as in the text. When the bromide ion is doped with polyaniline, 0.1 M KCl is used instead of 0.1 M KBr as the electroplating solution. As shown in the Fig.1, no significant redox peak is produced During the scanning process. The oxidation peak of 0.3 is formed by the deposition of CI- on the silver wire to form AgCl. It is proved that Br- is involved in the doping process.

Figure 1. Doped polyaniline in 0.1 M KCl Solution by cyclic voltammetry.\

Point 4: Figures 7 and 8 are interchanged. Be careful.

Response 4: This is an unforgivable mistake. Thank you for giving me this opportunity to amend. The order of Fig.7 and Fig.8 has been corrected.

Point 5: I miss statistical parameters, such as a coefficient of variation (CV (%)) to evaluate the repeatability (the same sensor, calibrate different times) and reproducibility (different sensors) of this new ISE.

Response 5: We have tested the reproducibility, repeatability and lifespan of the electrode, but it was deleted due to the length of the article. In Chapter 3, we add a section (3.7) for further explanation.

Point 6: The authors claim in the Abstract and in the Conclusions that the developed ISE has a long lifetime of more than 3 months… but there is no experimental data to prove it! Please, provide results to the main text.

Response 6: The lifespan of the Br-ISE experimental data was added to 3.7 section.

I am very grateful to you for giving me the opportunity to amend it. Please feel free to give me suggestions on how to do things better.

Reviewer 2 Report

Dear Editor,

The manuscript “Doped Polyaniline coated Nano-Ag electrode for rapid 3 in-situ detection of bromide in seawater” by Q. Wang et al. discuss important issue of bromine detection using potentiometric technique. Still, I cannot agree with publishing this manuscript in the present form. In the current form it should be rejected. However, the manuscript is rather interesting and shows good prospects for real applications, and I would recommend authors to revise it thoroughly.

English should be corrected.  Here are the suggestions:

1. Abstracts should be revised. Some information is too excessive for the abstracts (lines 21-22 might be removed as well as lines 22-23 – it is a repeat of what is mentioned earlier). Line 17. There is an extra dot, remove it. 

2. Introduction needs to be thoroughly re-written. There are lots of unclear sentences. E.g. what means “developed for a long history”? I would recommend re-phrase the sentence (lines 38-39). What means “relatively mature” (re-phrase, please). Line 42-43 – re-phrase the sentence (what means “liquid leakage interferes”)? 

Line 47-48: re-phrase the sentence. 

Line 51: authors did not mention and did not discussed inorganic membranes before. Please, comment.

Line 54: what means “applicable to seawater”? (Does it mean “applicable to detect bromine ions in seawater”?) Correct, please. 

Line 66-67. Please, remove the sentence and the table. There are no needs to put results in introductory part. 

3. Line 90. Preparation of nano silver is not carefully discussed. Authors should provide the references for the shown approach, and put more details.

4. Lines 99-101. A reference is required.  5. Line 103. Authors should separate “0.1M”.

6. The authors discuss the mechanism of electrode formation and present a schematic illustration in paragraph 2.3. Is there justification for the proposed mechanism? Reference?

7. Line 115. The sentence is not clear. Change it.

8. Figures 2, 3. Please indicate cycle number (use color)?

9. Line 140. It is not possible to estimate particle size from figure 4.

10. Line 142. Re-phrase “(roughness in nanoscale)”.

11. Line 144. How it was measured? (That the bond between the two layers reduces the interlayer conductive barrier). Or in other words, it looks rather speculative. I would also recommend to omit “between the two layers”. Provide reference.

12. Line 147. Correct English “is appears”.

13. Line 153. Where is R1? it is not logical to show R2 first.

14. Line 156-157. Authors should provide reference to this statement.

15. Linde 159. Please, re-phrase “deposited PANI in a mixed solution of …”

16. Line 174. Please, omit word “previous”.

17. Line 180. There is an extra dot. Remove it.

18. Lines 182-184. Interpretation is rather questionable. Did you account for solution resistance? Usually, double layer capacitance is in parallel with charge-transfer resistance. See Randles circuit

Moreover, solution resitance might be coupled with electrode resistance?

Nyquist plot should be corrected. One should use the same scale bar (it is quadratic). Otherwise, the impedance explanation might be not correct.

Line 185. Fig. 7 and Fig. 8 should be swapped.

Line 186. “Resistence” should be corrected to “resistance”.

Lines 192-194. This statement should be proved.

Line 202. How these values were calculated?

Line 205 (figure 7). Authors should explain what NO.1 and NO.2 means in text. The figure itself is not explained.

Line 210. Authors should correct indexes. (10-6 M – 10-1M).

Line 223 (figure 9). Time scale should be corrected. It looks like a row of numbers.

Line 225. Authors should correct indexes (10-3).

Line 238. I would recommend to use 10-3 as in the rest text instead of 1x10-3.

Line 299-300. This sentence is a bit speculative. Revise it.

Author Response

Point 1: Abstracts should be revised. Some information is too excessive for the abstracts (lines 21-22 might be removed as well as lines 22-23 – it is a repeat of what is mentioned earlier). Line 17. There is an extra dot, remove it.

Response 1: Abstracts has been revised, and lines 21-22 and an extra dot in line 17 have been removed.

Point 2: Introduction needs to be thoroughly re-written. There are lots of unclear sentences. E.g. what means “developed for a long history”? I would recommend re-phrase the sentence (lines 38-39). What means “relatively mature” (re-phrase, please). Line 42-43 – re-phrase the sentence (what means “liquid leakage interferes”)?

Line 47-48: re-phrase the sentence.

Line 51: authors did not mention and did not discussed inorganic membranes before. Please, comment.

Line 54: what means “applicable to seawater”? (Does it mean “applicable to detect bromine ions in seawater”?) Correct, please.

Line 66-67. Please, remove the sentence and the table. There are no needs to put results in introductory part.

Response 2: In order to make the text clearer, we have rewritten the introduction and used MDPI's professional English editing service. In addition, Line 42-43, Line 47-48, have re-phrased and we discussed inorganic membranes in Line 51.

For Line 54, we have re-phrased the sentence.

For Line 66-67, We have removed the Table 1 and the related comments to the Conclusions.

Point 3: Line 90. Preparation of nano silver is not carefully discussed. Authors should provide the references for the shown approach, and put more details.

Response 3: We have rewritten the preparation process of nano silver and added corresponding preparation principles and references.

Point 4: Lines 99-101. A reference is required.

Response 4: A reference was added in in line 119.

Point 5: Line 103. Authors should separate “0.1M”.

Response 5: We have separated “0.1M”.

Point 6: The authors discuss the mechanism of electrode formation and present a schematic illustration in paragraph 2.3. Is there justification for the proposed mechanism? Reference?

Response 6:We added references [42-43] in line 130 and explanations for different peaks. In addition, because polyaniline doping and cyclic voltammetry are complex, a supplementary experiment proves that the peak in Fig.3 is indeed generated by doping from the opposite side. The other steps are the same as in the text. When the bromide ion is doped with polyaniline, 0.1 M KCl is used instead of 0.1 M KBr as the electroplating solution. As shown in the Fig.1, no significant redox peak is produced During the scanning process. The oxidation peak of 0.3V is formed by the deposition of CI- on the silver wire to form AgCl. It is proved that Br- is involved in the doping process.

(I am very sorry, the figure cannot be uploaded. Please review it in the word version.)

Point 7: Line 115. The sentence is not clear. Change it.

Response 7: Line 115. The sentence has been revised.

Point 8: Figures 2, 3. Please indicate cycle number (use color)?

Response 8: We have indicated cycle number by colors in Fig.2 and Fig.3.

Point 9: Line 140. It is not possible to estimate particle size from figure 4.

Response 9In Figure 4b, the scale is 1μm, which can determine the size of silver particles to be nanometer. The specific size of silver particles in this paper is estimated and has been deleted.

Point 10Line 142. Re-phrase “(roughness in nanoscale)”.

Response 10: We have deleted it in line 142.

Point 11: Line 144. How it was measured? (That the bond between the two layers reduces the interlayer conductive barrier). Or in other words, it looks rather speculative. I would also recommend to omit “between the two layers”. Provide reference.

Response 11: There are many reasons for the low impedance of the electrode. We believe that polyaniline is synthesized in situ on the surface of the substrate, so there is no obvious conduction barrier between the substrate and the substrate. There is no reliable evidence in this paper. Therefore, we have deleted the polyaniline according to the reviewer's suggestion.

Point 12: Line 147. Correct English “is appears”.

Response 12: We have corrected it.

Point 13: Line 153. Where is R1? it is not logical to show R2 first.

Response 13: We modified the equivalent circuit diagram in Figure 7, so R1 is removed here.

Point 14: Line 156-157. Authors should provide reference to this statement.

Response 14: We have added a reference to explain it.

Point 15: Linde 159. Please, re-phrase “deposited PANI in a mixed solution of …”

Response 15: We have re-phrase this sentence.

Point 16: Line 174. Please, omit word “previous”.

Response 16: We have omitted previous text.

Point 17: Line 180. There is an extra dot. Remove it.

Response 17: We have removed an extra dot.

Point 18: Lines 182-184. Interpretation is rather questionable. Did you account for solution resistance? Usually, double layer capacitance is in parallel with charge-transfer resistance. See Randles circuit.

Response 18We have consulted the relevant literature. The explanation of equivalent circuit diagram in the manuscript is indeed wrong. It has been corrected in Fig.7. Rs is solution resistance and Rct is charge transfer resistance. In addition, according to the suggestion of another reviewer, because the scale of the three impedance diagrams is different, the datas of Fig.7 (a) and (b) have been deleted. Impedance part focuses on the impedance of the whole electrode, but also in order to save the length of an article.

Point 19: Line 185. Fig. 7 and Fig. 8 should be swapped.

Response 19: I am so sorry for it, we have corrected it.

Point 20: Line 186. “Resistence” should be corrected to “resistance”.

Response 20: We have corrected it.

Point 21: Lines 192-194. This statement should be

proved.

Response 21: Our electrode impedance belongs to the typical Semi-infinite diffusion model. The line with diffusion impedance of 45 degrees is formed by the diffusion of charge carrier in electrolyte.

Point 22: Line 202. How these values were calculated?

Response 22After the electrode is calibrated in 10-6M-10-1M KBr solution, the Nernst equation, including the slope and linear correlation coefficient, can be fitted. The Nernst slopes of the first and second electrodes are respectively: 57.28 mV/decade and 58.82 mV/decade.

The stability of the electrode is tested in Fig. 10. The difference between the maximum potential and the minimum potential is the potential drift of the electrode within 8 hours.

Point 23: Line 205 (figure 8). Authors should explain what NO.1 and NO Moreover, solution resitance might be coupled with electrode resistance?

Response 23: We have corrected the order of Fig.7 and Fig.8, so NO.1 and NO.2 are only the labels of the electrodes.

Point 24: Nyquist plot should be corrected. One should use the same scale bar (it is quadratic). Otherwise, the impedance explanation might be not correct.

Response 24: The data of Fig.7 (a) and Fig.7 (b) have been deleted. EIS part focuses on the impedance of the whole electrode. We have consulted the relevant literature. The explanation of equivalent circuit diagram in the manuscript is indeed wrong. It has been corrected in Fig.7: Rs is solution resistance and Rct is charge transfer resistance.

Point 25: Line 210. Authors should correct indexes. (10-6 M – 10-1M).

Line 223 (figure 9). Time scale should be corrected. It looks like a row of numbers.

Line 225. Authors should correct indexes (10-3).

Line 238. I would recommend to use 10-3 as in the rest text instead of 1x10-3.

Line 299-300. This sentence is a bit speculative. Revise it.

Response 25: We have corrected these points in manuscript.

I am very grateful to you for giving me the opportunity to amend it. Please feel free to give me suggestions on how to do things better.

Round 2

Reviewer 1 Report

I am quite satisfied with the changes made by the authors. The manuscript can be accepted in the present form.

Author Response

Dear Reviewer,

Thank you very much for all your pertinent suggestions for this manuscript. We have corrected the errors and other inappropriate textual expressions in manuscript according to the reviewer’s recommendations. In addition, my manuscript checked by my native english speaking colleague. The modified part of the manuscript has been marked.

Reviewer 2 Report

Dear Editor, there are some suggestions to authors to revise the manuscript.

Line 15. Instead of “the morphologies” use “the morphology”

Line 19. Instead of “performed” use “possessed”

Line 28-29. The sentence is unclear.

Line 87. Remove word “electrode”.

Line 100,99. Please, replace word certain with 0.1 M (from line 100).

Line 104. Not “cycle” but “cyclic”.

Line 105. Use “cycles” not “circles”

Line 108. Remove “response specific”.

Line 151. Right word is “impedance”.

Line 152-153. “is appears” is not correct. Check English.

Line 192. Remove “(c)”.

Fugure 9. Change time scale.  The numbers are too close to each other.

Line 227-228. Check. It is unclear.

Line 245. It is Table 1. The same for line 246.

Lines 251-256. Use the word “reproducibility” instead of “reproductivity”

Line 304. Use word “micrographs” instead of “photographs”.

Place Table 4 and corresponding text from conclusions before conclusions.

References 23, 40, 44 – check. There are some typesetting errors.

Generally, the manuscript needs serious English editing and text editing what I have not mentioned above. Therefore, it is a full responsibility of the Editor in case of acceptance of this work.

Author Response

Point 1: Line 15. Instead of “the morphologies” use “the morphology”

Line 19. Instead of “performed” use “possessed”

Line 28-29. The sentence is unclear.

Line 87. Remove word “electrode”.

Line 100, 99. Please, replace word certain with 0.1 M (from line 100).

Line 104. Not “cycle” but “cyclic”.

Line 105. Use “cycles” not “circles”

Line 108. Remove “response specific”.

Line 151. Right word is “impedance”.

Line 152-153. “is appears” is not correct. Check English.

Line 192. Remove “(c)”.

Fugure 9. Change time scale. The numbers are too close to each other.

Line 227-228. Check. It is unclear.

Line 245. It is Table 1. The same for line 246.

Lines 251-256. Use the word “reproducibility” instead of “reproductivity”

Line 304. Use word “micrographs” instead of “photographs”.

Place Table 4 and corresponding text from conclusions before conclusions.

References 23, 40, 44 – check. There are some typesetting errors.

Response 1: We have corrected the errors and other inappropriate textual expressions

in manuscript according to the reviewer’s recommendations. In addition, my manuscript checked by my native English speaking colleague. The modified part of the manuscript has been marked.